# The Impact of Climate Change Risks on Residential Consumption in China: Evidence from ARMAX Modeling and Granger Causality Analysis

**DOI:** 10.3390/ijerph191912088

**Published:** 2022-09-24

**Authors:** Miaomiao Niu, Guohao Li

**Affiliations:** 1School of Politics and Public Administration, Zhengzhou University, Zhengzhou 450001, China; 2Management School, Zhengzhou University, Zhengzhou 450001, China; 3Center for Energy, Environment & Economy Research, Zhengzhou University, Zhengzhou 450001, China

**Keywords:** climate change risks, residential consumption, ARMAX, granger causality

## Abstract

Estimating the impact of climate change risks on residential consumption is one of the important elements of climate risk management, but there is too little research on it. This paper investigates the impact of climate change risks on residential consumption and the heterogeneous effects of different climate risk types in China by an ARMAX model and examines the Granger causality between them. Empirical results based on monthly data from January 2016 to January 2019 suggest a significant positive effect of climate change risks on residential consumption, but with a three-month lag period. If the climate risk index increases by 1 unit, residential consumption will increase by 1.29% after three months. Additionally, the impact of climate change risks on residential consumption in China mainly comes from drought, waterlogging by rain, and high temperature, whereas the impact of typhoons and cryogenic freezing is not significant. Finally, we confirmed the existence of Granger-causality running from climate change risks to residential consumption. Our findings establish the linkage between climate change risks and residential consumption and have some practical implications for the government in tackling climate change risks.

## 1. Introduction

Since the industrial revolution, the global average temperature has continued to rise. The World Meteorological Organization (WMO) pointed out in the latest edition of *The State of the Global Climate* that the global average surface temperature in 2020 has increased by 1.2 °C compared with the pre-industrial period [1]. In June 2021, American scientists’ latest detected concentration of carbon dioxide in the earth’s atmosphere reached 419.13 parts per million, breaking the highest record in human history. Rising greenhouse gas concentrations are pushing global temperatures to increasingly dangerous levels. Global warming has increased the frequency and intensity of extreme climate events and affected the global socio-economic system, becoming one of the world’s biggest risks [2].

China is a sensitive and important area of global climate change. According to the China Climate Bulletin [3], from 1951 to 2018, the annual average temperature in China increased by 0.24 °C every decade, and the warming rate was higher than the global average in the same period. The extremely heavy rainfall and high-temperature events have clearly increased since the 1990s. China’s coastal sea level showed a fluctuating upward trend from 1980 to 2017. In 2017, the coastal sea level was 58 mm higher than the average level in 1993–2011, the fourth-highest level since 1980. At the same time, as the largest developing country, China is one of the fastest-growing economies globally. In recent decades, the consumption capacity of China’s residents has increased greatly. However, it is not clear whether residential consumption has been affected by climate change risks. Clarifying this issue may be highly significant to climate change risks management in China, especially after the Chinese government proposed to achieve carbon neutrality by 2060.

This study uses the ARMAX model to accurately model the relationship between residential consumption and climate change risks to investigate the impact of climate change risks on residential consumption. At the same time, we analyze the heterogeneous impact of different types of climate risks on residential consumption. In addition, the Granger causality between residential consumption and climate change risks is tested. To the extent of our knowledge, no study has investigated these. Our study is the first to establish the temporal linkage between climate change risks and residential consumption, widening the research boundary of climate risk management to the field of residential consumption.

The remainder of this paper is organized as follows. Section 2 reviews the literature relevant to this study. Section 3 describes the details of methods used in the empirical analysis, and the data source. Section 4 presents the empirical results and analysis. Section 5 presents the discussion. Section 6 concludes the paper with some key policy implications.

## 2. Literature Review

### 2.1. Impact of Climate Change Risks on Social-Economic Systems

Climate change risks are the potential adverse effects of climate change on natural and socio-economic systems due to changes in the climate system caused by natural and human disturbances [4]. The socio-economic impacts of climate change risks are increasing. Agricultural production [5,6] and human health [7,8] are directly affected by extreme climate events and are very sensitive to climate change risks. Tourism [9,10] and mining [11] industries, closely related to natural resources, are undergoing great changes due to climate risks. Industries that rely less on the natural environment are also affected by climate risks. For example, Huang, et al. [12] found the negative impact of climate risk on firm performance. Yang and Tang [13] suggested that high temperatures can lead to higher local fiscal stress, which further contributes to regional inequality. In general, climate risks affect the human socio-economic system in various aspects.

In terms of consumption, a noticeable impact of climate change risks is on energy consumption, especially electricity, which is also being focused on recently. The frequent occurrence of climate warming and extreme heatwaves has influenced the cooling load in summer and heating load in winter, leading to an increase in residential electricity consumption [14]. In the case of rising temperatures, residents will use air-conditioning more frequently, increasing electricity demand. Reyna and Chester [15] found that the residential electricity demand in Los Angeles may increase by 41–87% between 2020 and 2060. Alvi, et al. [16] showed that electricity consumption would increase in both the short and long term under climatic changes. This phenomenon has also occurred in China. Li, et al. [17] indicated that electricity consumption would increase by 7.9% and the annual peak electricity consumption would increase by 36.1% for every 1 °C increase in annual global mean surface temperature in the Yangtze River Delta. Zhang, et al. [18] found the different impacts of temperature fluctuation on electricity consumption between South and North China, and winter and summer. However, too little research has been done on the impact of climate change risks on residential consumption.

### 2.2. Time-Series Modeling Methods for Residential Consumption

Residential consumption has typical time-series characteristics [19], which means that current observations may be related to historical observations. The key to time-series analysis is to choose an appropriate model; if the model is not properly selected, the estimation results will be inaccurate. Since there is usually more than one factor influencing residential consumption, a multivariate time-series model is necessary. Commonly used multivariate time-series models include the dynamic-factor (DF) model, multivariate generalized autoregressive conditional heteroscedasticity (MGARCH) model, vector-autoregressive (VAR) model, and multivariate-autoregressive moving-average model (ARMAX). Among them, the DF model is good at processing high dimensional data sets, specifically extracting common factors from hundreds of variables [20,21]. The advantage of the MGARCH model is that it can accurately simulate the volatility of time series, so it is widely used in financial markets with high volatility characteristics [22,23]. The data involved in this study do not have the characteristics of high dimensionality and high-frequency volatility, so it is not suitable to use the DF and MGARCH models. The VAR model is a systematic forecasting method for interrelated variables [24,25,26]. Each variable in the model is explained by its lag value and the current value and past value of the remaining variables, but it lacks the consideration of the lag of the random disturbance term (i.e., moving average term). The VAR model cannot be rashly used until it is determined that the variable is not related to the moving average term. In comparison, the ARMAX model has a more flexible modeling mechanism and better compatibility in data characteristics.

The ARMAX model is an extension of the auto-regressive moving-average (ARMA) model with explanatory exogenous variables *X*. The ARMA model only explains the variable by its past values and stochastic error terms and does not consider other factors fully. The ARMAX model introduces other exogenous variables *X* into the ARMA model to enhance the model’s response to external factors, which can effectively improve the model’s estimation accuracy. The ARMAX model has been widely used in the research of time-series prediction and influencing factors analysis. For example, Lim, et al. [27] used the ARMAX model to estimate the income elasticity of Japanese residents traveling to New Zealand and Taiwan based on quarterly data from January 1980 to February 2004. Nitka and Burnecki [28] analyzed the relationship between the sunspot numbers and the average monthly precipitation measured at meteorological stations in the US employing the ARMAX model and found that the errors are essentially lower for the ARMAX model. Maddison [29] used the ARMAX modeling method to analyze the temporal relationship between the number of hospitalized people and air pollution in London by using daily data. Hossain, et al. [30] examined the impact of weather on COVID-19 transmission for the first time in five south Asian countries by an integrated ARMAX model. ARMAX can also be combined with other time-series models to expand the performance and application range of the model [31,32,33].

In addition to establishing the precise relationship between residential consumption and climate change risks, this study will also explore the causal relationship between them, which will help to strengthen the understanding of the impact of climate change risks on residential consumption. For the causality analysis of time series, the Granger causality test is the first choice. The Granger causality test is a classical method to test whether a group of time-series variables have causality [34]. Its basic principle is as follows: If *x* is the cause of *y*, but *y* is not the cause of *x*, the past value of *x* can help predict the future value of *y*, but the past value of *y* cannot help predict the future value of *x*. From this perspective, Granger causality reflects the prediction ability of one variable to another, rather than the real causality. Nevertheless, it is still the most effective method for the time-series causality test [35,36] and is widely used in the field of climate change economics [37,38].

## 3. Methodology and Data

This paper first models the impact of climate change risks on residential consumption, and then examines the Granger causality between them. The main methods used include the ARMAX model and the Granger causality test.

### 3.1. ARMAX Model

Given that the ARMAX model has a flexible modeling mechanism and good compatibility in data characteristics, this study adopts the ARMAX model to estimate the impact of climate change risks on residential consumption.

The auto-regression (AR) and moving-average (MA) models are the two most common basic univariate time-series models. That time-series *y_t_* is an AR (*p*) process represents that *y_t_* is related to its *p*-order lags. Similarly, that *y_t_* is an MA (*q*) process represents that *y_t_* is related to *q*-order lags of white noise. ARMA (*p*, *q*) model is the combination of the AR (*p*) model and MA (*q*) model, and it introduces exogenous variables *X* into ARMA (*p*, *q*). The ARMAX model with *k* exogenous variables can be expressed as follows [39]:(1)$$\alpha_{0}{+\alpha (L)y}_{t}=\overset{k}{\underset{i=1}{\sum }}\eta_{i}{(L)x}_{it}\;{+\theta (L)\varepsilon }_{t}\;$$
where it is an endogenous variable, *x_it_* is the *i*-th exogenous variable, and $\varepsilon_{t}$ is a white noise that is generally assumed to be independent, identically distributed variables and satisfy the standard normal distribution. *L* is the lag operator, which satisfies *L^p^*·*y_t_* = *y_t-p_* and *L^p^*·*L^q^* = *L^p+q^*
α(L), $\eta_{i}\left(L\right)$ and θ(L) denote the lag polynomial of *y_t_*, *x_it_* and $\varepsilon_{t}$, respectively, and can be calculated as
(2)$$\;\alpha (L)=1\;-\overset{p}{\underset{j=1}{\sum }}\alpha_{j}L^{j}\;$$
(3)$$\;\theta (L)=1\;-\overset{q}{\underset{j=1}{\sum }}\theta_{j}L^{j}\;$$
(4)$$\eta_{i}{(L)=\eta }_{i0}-\overset{s_{i}}{\underset{j=1}{\sum }}\eta_{ij}L^{j}\;$$
where $\left\{\alpha_{j}\right\}$, $\left\{\theta_{j}\right\}$ and $\left\{\eta_{ij}\right\}$ are the coefficients of AR, MA and the exogenous variables, respectively. *s_i_* is the maximum lag order of the *i*-th exogenous variable.

The construction of the ARMAX model is an iterative process, including the following steps [40]:(1)The unit root test is performed on all variables to determine the stationarity of the variable series. If all variables are stationary time series, the next steps can be proceeded directly; otherwise, the differencing operation must be performed on all variables until all variables are stationary at the same integrated order.(2)Identify the structure of the ARMAX model. Specifically, first determine the maximum lag order *p* and *q* of the AR term and MA term in ARMAX (*p*, *q*), and then determine the maximum lag order of independent variable *X*.(3)Estimate the coefficients of the ARMAX model and determine the optimal ARMAX model according to the statistical significance of the estimated coefficient, quality of fit and information value.(4)Conduct diagnostic tests of the estimated model to ensure its applicability. The time-series residuals must meet the white noise assumptions; otherwise, the residual series contains additional information that might be used by a more complex model.

If the ARMAX model is finally used for prediction, the next step should be to predict and compare the prediction performance with other models. However, the purpose of this study is not a prediction, so it only includes the above four steps.

### 3.2. Granger Causality Test

To explore the causal relationship between residential consumption and climate change risks, this study conducts a Granger causality test, which is one of the most effective methods for the time-series causality test.

The Granger causality test is usually examined by estimating the VAR model or error correction model (ECM) model. Specially, if variables are known to be stationary (i.e., zero-order integrated, denoted by I(0)) or first-order integrated (denoted by I(1)), one can conduct a VAR for the levels or first-order differences of the variables, respectively; if the variables are known to be cointegrated, one can conduct an ECM. Hence the Granger causality test should first test the stationarity of variables. However, as Toda [41] pointed out, the pre-test is very sensitive to correctly identifying the integration order and selecting the lag length. Hence there may be severe pre-test bias in causal inference in ECM. If the system contains unit roots, standard Wald statistics based on ordinary least-squares (OLS) estimation of level VAR model for testing coefficient restrictions have non-standard asymptotic distributions that may involve nuisance parameters. For this problem, the augmented VAR model proposed by Toda and Yamamoto [42] (thereafter TY) can overcome the above-mentioned defects because it can be applied to any level of integration. TY modifies the Wald test in the augmented VAR model, and there is no need to pre-test the cointegration of time series. As proved by Toda and Yamamoto [42], the modified Wald statistic converges to the chi-square random variables in distribution, whether it is stationary or not. In addition, the TY procedure involves level VAR, so there is no loss of information due to differencing [25]. Recent research on the TY procedure can be referred to Saint Akadiri, et al. [43]. The TY procedure steps are as follows:(1)Perform unit root test to find the maximal order of integration *d* of variables.(2)Determine the optimal lag length *p* of VAR, but since the real lag length *p* is rarely known in practice, we can evaluate it through several criteria.(3)Estimate the augmented VAR (*p* + *d*) model:
(5)$$V_{t}{=\alpha +\beta }_{1}V_{t-1}{+\beta }_{2}V_{t-2}+\cdots {+\beta }_{p}V_{t-p}+\cdots {+\beta }_{p+d}V_{t-p-d}{+\varepsilon }_{t}\;$$
where *α* is constant vector, *β_t_* is coefficient matrix and *ε_t_* is the residual of white noise.(4)Conduct diagnostic checking to verify the robustness of the augmented VAR (*p*+*d*). The Wald test is performed on the first *p* parameters in the augmented VAR (*p*+*d*) model, instead of on all parameters. The statistics obey the asymptotic chi-square distribution of *p* degrees of freedom.

### 3.3. Variables and Data Source

This study aims to investigate the impact of climate change risks on residential consumption. This obviously means that the dependent variable and the core independent variable are residential consumption and climate change risks, respectively. Residential consumption may also be affected by other factors, one of which, typically, is the price level of consumer goods. According to the consumer demand theory, the demand of consumers for general commodities and their prices change in the opposite direction. In general, a higher price level of consumer goods will inhibit residential consumption. Therefore, this study takes the price level of consumer goods as the exogenous variable of the ARMAX model. Some scholars have noticed that house price fluctuations will have a huge impact on residential consumption, and house purchase costs will squeeze households’ consumption of other commodities [44]. However, in China’s official statistics, the consumer price index representing the price level of consumer goods already includes house prices, so this study will no longer take house prices as a separate exogenous variable. In addition, as one of the important sources of residential income, the rate of return of the financial market may also affect residential consumption, but the direction of the impact is uncertain. The description and measurement of all variables in this study are as follows.

(1) Residential consumption. Residential consumption is measured by the retail sales of social consumer goods (URS), which is the most direct indicator of consumption demand in various consumption-related statistics. URS is the total amount of consumer goods directly sold by various industries of the national economy to residents and social groups. URS is the actual consumption that has been generated.

(2) Climate change risks. We use the climate risk index (CRI) that is jointly issued by Caixin Insight Group and National Climate Center to measure the climate change risks. CRI is a quantitative assessment of single or comprehensive climate disaster risk based on historical climate data and future climate prediction results. The comprehensive CRI can be divided into the index for waterlogging by rain (WRI), drought index (DI), typhoon index (TI), high-temperature index (HI), and cryogenic freezing index (CFI). The construction method of the climate risk index can be seen in Wang, et al. [45].

(3) Price of consumer goods. The price level of consumer goods is measured by the consumer price index (CPI). The CPI is weighted by the prices of a group of representative goods and services, which can well reflect the changes in the general price level of goods and services purchased by households. The CPI statistical survey is the final price of social goods and services.

(4) Return ratio of the financial market. Because stocks have become the most common investment tool for Chinese residents [46], we use the stock return (SR) to measure the return ratio of the financial market.

The data used in this study are from the following three sources: the data of URS and CPI are obtained from the National Bureau of Statistics of China (The website is https://data.stats.gov.cn/easyquery.htm?cn=A01); the CRI series data are from the monthly report of the China climate index (The website is http://www.ncc-cma.net/climate-index/); the data of SR are obtained from the China Stock Market and Accounting Research Database (CSMAR) (The website is http://www.gtarsc.com/). All data are monthly. Because the CRI data are only available from January 2016 to January 2019, our analysis is confined to this period. In addition, to avoid the big difference in the order of magnitude among variables, URS is converted to the form of natural logarithms, that is, lnURS. The descriptive statistics of the variables are summarized in Table 1.

## 4. Empirical Results

### 4.1. ARMAX Model Identification and Estimation

Before identifying the structure of the ARMAX model, it was necessary to select appropriate exogenous variables. Although we had preliminarily identified two non-core exogenous variables, they were not necessarily conducive to model estimation. Here, we inspected the multicollinearity and stationarity to filter variables.

Severe multicollinearity can impair the performance of the model. We calculated the variance inflation factor (VIF) values of independent variables, which were 1.31 (*CRI*), 4.31 (*CPI*) and 1.06 (*SR*), respectively. Since the VIF values of all variables were less than 5, the multicollinearity problem could be rejected.

The ARMAX model was only suitable for stationary time series, so the unit root test was needed before properly identifying the ARMAX model. We conducted four different unit root tests, namely the augmented Dickey-Fuller (ADF), the Dickey-Fuller GLS (DF-GLS), the Phillips-Perron (PP) and the Kwiatkowski-Phillips-Schmidt-Shin (KPSS). The results of the unit root tests are reported in Table 2. According to Table 2, the variables *lnURS*, *CRI*, *CPI* and *SR* were stationary in at least one case with or without trend. Therefore, the next steps could be conducted.

The second step was to determine the model structure, that is, to determine the orders *p* and *q* in the ARMA (*p*, *q*). According to the Box–Jenkins’ method [47], one could calculate the series autocorrelation coefficient (AC) and partial autocorrelation coefficient (PAC) to determine the orders *p* and *q* of the ARMA models. A pure MA (*q*) process exhibits a cut-off in its AC function (ACF) for lags greater than *q*, and its PAC function (PACF) tails off to zero. On the other hand, the ACF of pure AR (*p*) tails off to zero, and its PACF exhibits a cut-off for lags greater than *p*. Besides, both the ACF and PACF of ARMA (*p*, *q*) tail off to zero. Figure 1 shows the ACF and PACF diagram of the dependent variable *LnURS*.

As can be seen from Figure 1, the ACF evidently tailed off to zero, and the PACF evidently presented a cut-off, which is completely in accordance with the condition of AR (*p*). Since the PAC in the first-order lag was significantly not 0 at the level of 5%, *p* was determined to be 1. The lag order of independents was determined by the significance of regression coefficients.

The next step was to determine the lag order of independents. Here we directly estimated the ARMAX model. The lag orders of independent variables and their coefficients were simultaneously determined according to the significance of regression coefficients. We used the stepwise elimination method of multiple linear regression to find the optimal ARMAX model. For parameter estimation techniques, referring to Li, Su and Shu [40], we used maximum-likelihood estimates. First, all independent variables and their *k*-order lag terms were included in the model. *k* is the maximum lag order of the independent variables that must be determined initially. Based on the lag order of the autoregression term of the dependent variable (that is, 1), we appropriately expanded the search range of the lag order of the independent variables, and thus we set all independent variables’ *k* to 3. Then, the term with the largest *p*-value in the t-test of coefficients was gradually eliminated. Finally, the optimal model could be obtained until the coefficients of all terms are significant at the level of 5%. Table 3 reports the regression process and results of the ARMAX model estimate.

The coefficients of the third-order lag terms of *CRI* (denoted as *CRI_t_*_−3_) were always significant in models (1)–(10), whereas the coefficients of *CPI_t_*_−3_ and AR(1) were gradually significant. All models performed well in terms of quality of fit, which were in the range of 0.86–0.90. In addition, we calculated the information value of the model and found that the AIC and BIC of the model (10) were the lowest. So far, model (10) seemed to be the best ARMAX model. Finally, to evaluate if the residual series was white noise, the Ljung-Box Q test was performed on the residual of the model (10). The Ljung-Box test P-value was 0.1054 and higher than 0.05, which did not show any autocorrelation in the residual of the model. Therefore, we could confidently determine that model (10) was the optimal ARMAX model. Accordingly, the relationship between climate change risks and residential consumption could be obtained as follows:*lnURS_t_* = 0.5562 · *lnURS_t_*_−1_ + 0.0129 · *CRI_t_*_−3_ + 0.0389 · *CPI_t_*_−3_ + 6.2742

### 4.2. Analysis on the Impact of Climate Risk on Residential Consumption

In the final determined model, the lag orders of the independent variables *CRI* and *CPI* were determined as 3. Another independent variable *SR* was removed because the coefficients of all its lag terms were not significant. This also showed that stock market volatility had no significant impact on household consumption, possibly for the reason, as pointed out by Xue [46], that the overall scale of China’s stock market is still very small, the proportion of stock assets in residential wealth is low, and the continuous fluctuation trend of stock market value is unstable, which affects market expectations. Therefore, the return ratio of the stock market iwas not enough to affect residential consumption.

Although the lag order of *CPI* was 3 in the final model, the lag order of *CPI* was not absolute. For example, in model (8), the coefficient of the one-order lag term of *CPI* was significant at 5% level. In fact, by replacing *CPI_t_*_−3_ in model (10) with *CPI_t_*_−1_, a robust estimation could also be obtained. However, the significance of *CPI_t_*_−1_ changes with the removal of some variables in the model. Therefore, it can be concluded that the price level of consumer goods can promote residential consumption, but the promotion effect will suffer interference from other factors.

The impact of climate change risks on residential consumption is significantly positive but has apparent hysteresis. The lag period is three months, which can also be intuitively seen in Figure 2. The reason may have two aspects. First, many residential consumption activities that are sensitive to climate change risks, such as home decoration and travel, are severely restricted and accumulated when the climate risk is high. Once the climate risk is mitigated, the demand for consumption is gradually released. On the other side, climate events will also cause great damage to agricultural production, resulting in the decline of agricultural products (especially fresh food) output and the rise of agricultural products’ prices, which will increase residential expenditure on agricultural products’ consumption during the climate events. After the mitigation of climate risk, the residential expenditure on agricultural products will decrease with the fall of prices and the recovery of production. This means that there is no time lag in the residential consumption of agricultural products. However, the residential expenditure on agricultural products is far less than those restricted consumption activities. Therefore, in general, the impact of residential consumption by climate risk is positive and lagging behind.

In addition, although we have confirmed a statistically significant positive effect on residential consumption, the effect is very small. The coefficient of *CRI_t_*_−3_ is 0.0129, which indicates that every one unit increase in the climate change risks index will increase residential consumption by 1.2984% after three months.

### 4.3. Heterogeneous Effects of Different Climate Risk Types

There are big differences between subdivided climate risks. From Figure 2, the main climate risks in China are waterlogging by rain and high temperature, which will reach a high-risk level every summer. The next is typhoon. However, the typhoon risk is highly uncertain. For example, the highest typhoon risk index in 2018 reached 10, but in 2017 it was only 2.65. It is followed by drought. The drought risk index in the sample period reached 5.85 only in September 2016, and in other months they were all lower than three. The last is cryogenic freezing, which mainly occurs in parts of the northeast and northwest, and the overall risk is relatively low.

Different types of climate risks may have different impacts on residential consumption. Using the form of the ARMAX model determined by basic regression, we replaced the core independent variable *CRI_t_*_−3_ with subdivided climate risks indices and performed the regression analysis again. Table 4 reports the regression result of the different impacts of different types of climate risks on residential consumption.

It can be seen from Table 4 that the coefficients of *WRI_t_*_−3_, *DI_t_*_−3_ and *HI_t_*_−3_ are significantly positive at 1% level, whereas the coefficients of *DI_t_*_−3_ and *TI_t_*_−3_ are not statistically significant. This result is essentially consistent with the duration and coverage of various types of climate change risks. Typhoons occur only along with the coastal provinces and usually last a short time, and thus have no obvious effect on national residential consumption. Cryogenic freezing lasted for a long time in northeast and northwest China, but is limited to partial regions, which often belong to low consumption areas. Therefore, the impact of cryogenic freezing on national residential consumption is also not significant. In summer, high-temperature weather generally occurs in most of China, waterlogging by rain is widely distributed in the south of China, and drought often occurs in the north. These three climate types tend to last for several months, so their impact on national residential consumption is significant.

### 4.4. Granger Causality Test of Climate Change Risks on Residential Consumption

Table 3 shows that *SR* had no significant impact on residential consumption. Therefore, the Granger causality test in this section only considered the interaction between *lnURS*, *CRI* and *CPI*. At the same time, the ARMAX model as finally identified only contained the AR term, which is internally consistent with the Granger causality test based on the VAR model.

The unit root test in Table 2 shows that the integration orders of *lnURS*, *CRI* and *CPI* are 0; thus we determined the maximal order of integration to be 0 (*d* = 0). Next, we determined the optimal lag length. We used five different criteria to determine the lag length, namely the likelihood ratio (LR) test, Akaike information criterion (AIC), Hannan–Quinn information criterion (HQIC) and Structural Bayesian information criterion (SBIC). The lag length selection results were reported in Table 5. LR and AIC suggested the optimum lag length of 4, although FPE, HQIC and SBIC pointed out the optimum lag length of 3. Here, we chose the latter (*p* = 3). Therefore, we estimated the augmented VAR (3) (*p* + *d* = 3) and conducted a series of diagnostic tests.

Firstly, the Wald test was carried out on the joint significance of the coefficients of each order of the three equations and all equations in the VAR (3) system. The results are shown in Table 6. Although some order coefficients of a single equation were not significant, the coefficients of each order for the whole VAR (3) system were highly significant. Further, Table 7 shows that the values of the adjusted R^2^ were rather high, and thus the explanatory power of all equations was robust. The Jarque-Bera test results showed that the residuals of all equations obeyed normal distribution. The Lagrange-Multiplier (LM) test results showed that the residual is white noise, and there was no autoregressive conditional heteroscedasticity. In addition, Figure 3 showed that all eigenvalues are within the unit circle, indicating that the VAR (3) system was stable.

These diagnostic test results showed the accuracy of the VAR (3) system. We proceeded with the Granger causality test, and the results are shown in Table 8. We found the existence of unidirectional Granger-causality running from *CRI* to *lnURS* and bidirectional Granger-causality between *CPI* and *lnURS*. This was consistent with our expectations. The climate change risks had indeed become a cause of changes in residential consumption. This result strengthened the credibility of the impact of climate change risks on residential consumption based on the ARMAX model.

## 5. Discussion

Residential consumption is an essential driver of economic growth. Past studies on the impact of climate change on the economic system focused on the production side, and only a few paid attention to the consumption side, mainly energy consumption. This paper investigates the impact of climate change risks on residential consumption using two time-series analysis techniques. Climate change risks have a lagging positive effect on residential consumption, which usually peaks three months after the high climate change risk. We confirmed the heterogeneity impact of climate change risks on residential consumption. In addition, the causal relationship between climate change risk and residential consumption was rigorously verified, strengthening the credibility of this paper’s conclusions.

Our findings are a beneficial complement to the existing literature. A recent study showed that temperature shocks have a direct negative impact on residential consumption [48], and several other studies found that extreme temperatures could reduce the outdoor activities of residents [49,50]. However, high temperature is only one dimension of climate change risks. The climate change risk in this paper is a more comprehensive indicator that includes high temperatures and disasters caused by climate change, such as waterlogging, drought, typhoon, and cryogenic freezing. In addition, this paper suggests that the positive impact of climate change risks on residential consumption has a lagged effect, as opposed to the current-period negative impact found in the existing literature. According to the methods and data used in this paper, it cannot be concluded that climate change risk has a significant negative impact on current-period residential consumption, although its regression coefficient is negative. Combined with the findings of existing literature, it can be inferred that the reason for the lagging positive impact of climate change risks on residential consumption may be that the climate change risks inhibit current-period residential consumption. After the climate change risks are mitigated, the inhibited consumption will be released, forming retaliatory consumption.

However, whether the lagging retaliatory consumption can completely offset the inhibited current-period consumption is uncertain; this is not addressed in this paper. If the lagging retaliatory consumption is insufficient to offset the inhibited current consumption, the damage to economic growth from climate change risks may be long-term. This issue will be focused on addressing in our future studies.

## 6. Conclusions and Policy Implications

Based on time-series data from January 2016 to January 2019, this paper investigates the effect of climate change risks on residential consumption by applying the ARMAX model that includes two additional exogenous variables. On this basis, we also tested the Granger causality between climate change risks and residential consumption. The conclusions of this paper are as follows.

Firstly, climate change risks have a significant lag effect on residential consumption, with a lag period of three months. Specifically, every increase of one unit in the climate risk index increases residential consumption by 1.29%.

Secondly, different types of climate risks may have different impacts on residential consumption. The impact of typhoons and cryogenic freezing on residential consumption is not statistically significant whereas waterlogging by rain, drought, and high temperature are the opposite.

Finally, we confirmed the existence of Granger-causality running from climate change risks to residential consumption. The Granger-causality from climate change risks to residential consumption is unidirectional and the Granger-causality between price of consumer goods and residential consumption is bidirectional.

The findings of this paper have the following implications for government decision-making. First, overall, the effect of climate risk on residential consumption is modest, so there is no need to take special measures to stimulate consumption after the occurrence of climate events. Second, during periods of high climate risks, some consumer goods are usually in short supply, and their price levels will rise in high-risk areas. Therefore, the government should reserve these materials in advance or allocate and transport them from low-risk areas.

Regarding follow-up research, the precise assessment and measurement of climate change risks is a question worth investigating, which is the basis for studying the impact of climate change risks on socio-economic systems. This study uses a readily available dataset from commercial institutions. But we believe that a systematic framework for measuring climate change risks is necessary. In addition, this paper shows the impact of climate change risks on residential consumption from a statistical perspective but fails to identify its impact mechanism. A feasible way of thinking in the future is to analyze the response of residents in the event of sudden climatic events from the perspective of consumer behavior, to explain how climate change risks affects the consumption behavior of residents.

## Figures and Tables

**Figure 1 ijerph-19-12088-f001:**
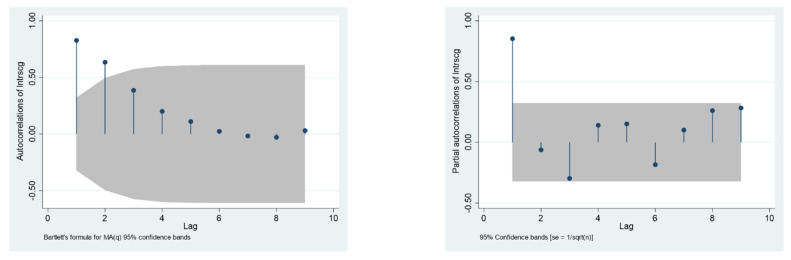
ACF and PACF diagram of *lnURS*.

**Figure 2 ijerph-19-12088-f002:**
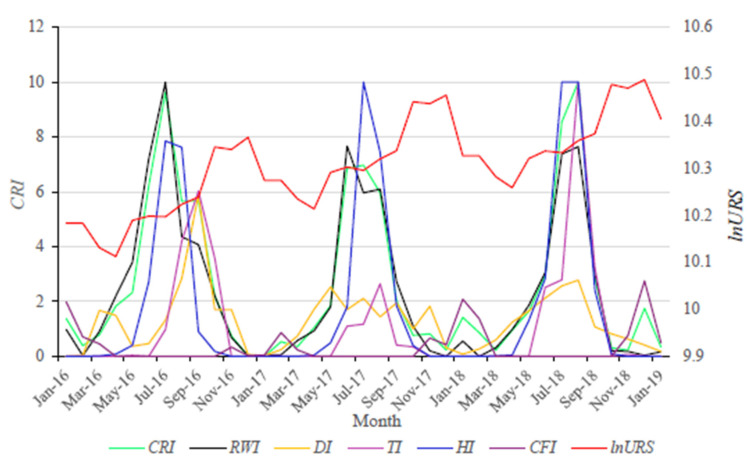
Monthly changes in residential consumption and comprehensive and subdivided climate risk index.

**Figure 3 ijerph-19-12088-f003:**
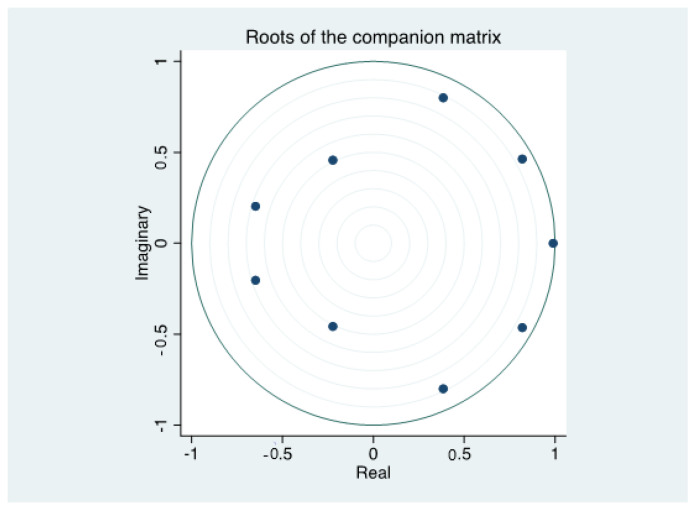
Stability test of VAR(3) system.

**Table 1 ijerph-19-12088-t001:** Descriptive statistics of the variables.

Variables	Mean	Std.	Min	Max	Observations
*lnURS*	10.3067	0.0973	10.1124	10.4883	37
*CRI*	2.6027	2.9004	0.0000	10.0000	37
*CPI*	4.6370	0.0158	4.6102	4.6657	37
*SR*	−0.0069	0.0537	−0.2265	0.1175	37

**Table 2 ijerph-19-12088-t002:** Unit root test results.

	Variables	ADF	DF-GLS	PP	KPSS
Intercept	*lnURS*	−1.735 ** (1)	−1.017 (1)	−1.757	0.883 ***
	*CRI*	−3.858 *** (1)	−3.680 *** (1)	−2.812 **	0.091
	*CPI*	−0.825 (0)	0.479 (1)	−0.721	1.8 ***
	*SR*	−6.886 *** (1)	−0.881 (3)	−9.231 ***	0.116
Intercept and trend	*lnURS*	−3.862 ** (2)	−3.983 *** (2)	−2.814 **	0.0888
	*CRI*	−3.837 ** (1)	−3.712 ** (1)	−2.770 **	0.081
	*CPI*	−4.145 *** (1)	−3.456 ** (1)	−3.251 **	0.0862
	*SR*	−9.334 *** (1)	−1.320 * (3)	−12.067 ***	0.101

Note: Superscripts ***, ** and * indicate statistical significance at 1%, 5% and 10% respectively. Lag lengths were determined via AIC and are in parentheses. The null hypothesis of all tests except KPSS is that the series had a unit root against the alternative of stationary. The null hypothesis of KPSS, on the contrary, was that the variable is stationary.

**Table 3 ijerph-19-12088-t003:** Process and results of ARMAX model estimation.

Models	(1)	(2)	(3)	(4)	(5)	(6)	(7)	(8)	(9)	(10)
Variables	*lnURS*	*lnURS*	*lnURS*	*lnURS*	*lnURS*	*lnURS*	*lnURS*	*lnURS*	*lnURS*	*lnURS*
*CRI_t_*	−0.0015	−0.0013	−0.0010	−0.0011	−0.0010					
	(0.0061)	(0.0055)	(0.0053)	(0.0053)	(0.0055)					
*CRI_t_* _−1_	0.0007	0.0006								
	(0.0090)	(0.0087)								
*CRI_t_* _−2_	−0.0010	−0.0012	−0.0010							
	(0.0087)	(0.0076)	(0.0070)							
*CRI_t_* _−3_	0.0131 **	0.0134 **	0.0135 **	0.0127 **	0.0122 ***	0.0124 ***	0.0118 ***	0.0118 ***	0.0119 ***	0.0129 ***
	(0.0056)	(0.0055)	(0.0055)	(0.0051)	(0.0046)	(0.0043)	(0.0043)	(0.0041)	(0.0036)	(0.0034)
*CPI_t_*	0.0062	0.0069	0.0075	0.0054						
	(0.0310)	(0.0310)	(0.0307)	(0.0293)						
*CPI_t_* _−1_	0.0250	0.0250	0.0250	0.0253	0.0289 *	0.0299 *	0.0294 *	0.0268 **	0.0131	
	(0.0210)	(0.0209)	(0.0208)	(0.0202)	(0.0169)	(0.0170)	(0.0173)	(0.0135)	(0.0143)	
*CPI_t_* _−2_	−0.0260	−0.0256	−0.0266	−0.0252	−0.0242	−0.0239	−0.0235	−0.0185		
	(0.0317)	(0.0204)	(0.0201)	(0.0201)	(0.0203)	(0.0206)	(0.0214)	(0.0197)		
*CPI_t_* _−3_	0.0386	0.0386	0.0390	0.0392 *	0.0395 **	0.0388 *	0.0377 *	0.0324 *	0.0299 **	0.0389 ***
	(0.0256)	(0.0246)	(0.0239)	(0.0233)	(0.0194)	(0.0199)	(0.0195)	(0.0191)	(0.0120)	(0.0079)
*SR_t_*	0.0779	0.1074	0.1164	0.0857	0.0639	0.0636				
	(0.4240)	(0.2971)	(0.2811)	(0.2798)	(0.2806)	(0.2777)				
*SR_t_* _−1_	0.2135	0.2556	0.2584	0.2377	0.2361	0.2342	0.1915			
	(0.6893)	(0.3094)	(0.3025)	(0.2978)	(0.2574)	(0.2558)	(0.2374)			
*SR_t_* _−2_	−0.0332									
	(0.7126)									
*SR_t_* _−3_	0.2349	0.2514	0.2481	0.2356	0.2266	0.2240	0.2114	0.1977		
	(0.3258)	(0.1931)	(0.1811)	(0.1841)	(0.1806)	(0.1790)	(0.1812)	(0.1860)		
*Constant*	5.7769 ***	5.6563 ***	5.6551 ***	5.6865 ***	5.7396 ***	5.6632 ***	5.7803 ***	6.0903 ***	5.8577 ***	6.2742 ***
	(2.1266)	(1.3701)	(1.3342)	(1.3916)	(1.5080)	(1.4778)	(1.5145)	(1.2276)	(0.9968)	(0.8192)
AR(1)	0.4416 *	0.4232 *	0.4162 *	0.4481 **	0.4908 ***	0.4915 ***	0.5208 ***	0.5161 ***	0.5633 ***	0.5562 ***
	(0.2421)	(0.2200)	(0.2181)	(0.2044)	(0.1583)	(0.1534)	(0.1378)	(0.1474)	(0.1349)	(0.1428)
Sigma	0.0386 ***	0.0387 ***	0.0387 ***	0.0387 ***	0.0387 ***	0.0388 ***	0.0388 ***	0.0396 ***	0.0419 ***	0.0425 ***
	(0.0059)	(0.0058)	(0.0057)	(0.0057)	(0.0052)	(0.0050)	(0.0050)	(0.0054)	(0.0061)	(0.0061)
*R* ^2^	0.8973	0.8966	0.8956	0.8943	0.8902	0.8901	0.8896	0.8802	0.8609	0.8698
*AIC*	−94.515	−96.508	−98.984	−100.424	−102.330	−104.240	−106.183	−106.771	−106.876	−107.890
*BIC*	−71.620	−75.139	−78.642	−82.108	−85.540	−88.976	−92.445	−94.560	−97.718	−100.258
*N*	34	34	34	34	34	34	34	34	34	34

Note: Standard errors are reported in parentheses. ***, **, * indicate statistical significance at 1%, 5%, 10% levels, respectively. The test of the variance against zero was one sided, and the two-sided confidence interval was truncated at zero.

**Table 4 ijerph-19-12088-t004:** Heterogeneous effects of different types of climate risks on residential consumption.

Models	(1)	(2)	(3)	(4)	(5)
Variables	*lnURS*	*lnURS*	*lnURS*	*lnURS*	*lnURS*
*WRI_t_* _−3_	0.0142 ***				
	(0.0039)				
*DI_t_* _−3_		0.0223 ***			
		(0.0084)			
*Ti_t_* _−3_			0.0050		
			(0.0066)		
*Hi_t_* _−3_				0.0112 ***	
				(0.0036)	
*CFI_t_* _−3_					−0.0261
					(0.0302)
*CPI_t_* _−3_	0.0414 ***	0.0298 **	0.0292 **	0.0364 ***	0.0287 *
	(0.0084)	(0.0123)	(0.0139)	(0.0081)	(0.0155)
*Con*	6.0131 ***	7.2177 ***	7.2986 ***	6.5412 ***	7.3613 ***
	(0.8619)	(1.2627)	(1.4268)	(0.8327)	(1.6053)
AR(1)	0.5115 ***	0.6996 ***	0.7170 ***	0.5927 ***	0.6907 ***
	(0.1601)	(0.1520)	(0.1386)	(0.1296)	(0.1525)
Sigma	0.0431 ***	0.0432 ***	0.0482 ***	0.0409 ***	0.0477 ***
	(0.0057)	(0.0063)	(0.0063)	(0.0070)	(0.0064)
*R* ^2^	0.8566	0.7786	0.7458	0.8683	0.7421
*N*	34	34	34	34	34

Note: Standard errors are reported in parentheses. ***, **, * indicate statistical significance at 1%, 5%, 10% levels, respectively. The test of the variance against zero is one sided, and the two-sided confidence interval is truncated at zero.

**Table 5 ijerph-19-12088-t005:** VAR lag length selection criteria.

Lag.	LL	LR	FPE	AIC	HQIC	SBIC
0	−96.1707		0.08182	6.01034	6.05612	6.14639
1	−30.5632	131.21	0.002657	2.57959	2.76269	3.12377
2	−12.6926	35.741	0.001577	2.04198	2.3624	2.9943
3	7.55517	40.496	0.00083 *	1.36029	1.81805 *	2.72075 *
4	16.78	18.45 *	0.000883	1.34677 *	1.94175	3.11527

Note: * marks the optimal lag length determined using the corresponding method.

**Table 6 ijerph-19-12088-t006:** Joint significance test results.

lag	*lnURS*	*CRI*	*CPI*	*All*
1	20.72926 ***	22.79644 ***	15.18021 ***	60.49401 ***
2	11.41464 **	1.264724	13.83375 ***	32.38024 ***
3	35.44091 ***	13.29102 ***	3.013248	57.87434 ***

Note: The statistics reported in this table are chi-square values. ***, **, * indicate statistical significance at 1%, 5%, 10% levels, respectively.

**Table 7 ijerph-19-12088-t007:** Test results of J-B test and ARCH LM.

Equations	Adj. R^2^	J-B Test	ARCH LM
*lnURS*	0.8625	0.881	1.423(3)
*CRI*	0.6478	0.460	8.274(4)
*CPI*	0.9532	1.574	2.253(2)

Note: The null hypothesis of J-B test is normality. The null hypothesis of ARCH LM is no ARCH up to the selected lag. Lag lengths are selected by SC and showed in parentheses. ***, **, * indicate statistical significance at 1%, 5%, 10% levels, respectively.

**Table 8 ijerph-19-12088-t008:** Granger causality test results.

Equations	*lnURS*	*CRI*	*CPI*	All	Direction of Causality
*lnURS*		17.813 ***	26.637 ***	52.928 ***	*CRI → lnURS*, *CPI → lnURS*,
*CRI*	5.011		3.371	5.206	
*CPI*	38.818 ***	20.050 *		46.882 ***	*lnURS → CPI*

Note: The statistics reported in this table are chi-square values. ***, **, * indicate statistical significance at 1%, 5%, 10% levels, respectively. The null hypothesis was that the independent variables were not the Granger cause of the corresponding dependent variable. “All” represents all independent variables.

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
