# Peer review of "The Impact of Climate Change Risks on Residential Consumption in China: Evidence from ARMAX Modeling and Granger Causality Analysis"

_ijerph, 2022, doi:10.3390/ijerph191912088_

Round 1
Reviewer 1 Report
The manuscript consists in a research on the impact of the climate change risks on the residential consumption in China. The paper is based on the ARMAX modeling and the Granger causality analysis.
The topic addressed by the manuscript is of high importance in the present context of significant climate changes with direct consequences for the economy of each country and for the consumption of each individual.
The paper provides a sound approach of the topic, but some aspects need further consideration:
a) Literature Review. The current version of the manuscript includes the literature review in the section ”1. Introduction”. In order to underline the different viewpoints of researchers, a distinct section of the manuscript can be dedicated to the literature review, immediately after a revised ”1. Introduction”.
b) Methodology: Research question(s). It would be useful to specify clearly the research question(s) at the beginning of the section ”Methodology and data”. Also, it would be necessary to explain the reasons for the use of the ARMAX modeling and of the Granger causality analysis in relation to the research question(s).
c) Implications for the research. The manuscript enumerates several policy implications. Nevertheless, the paper can benefit from the presentation of the implications for further research on the same topic. Examples of implications for future research can be provided.
This manuscript corresponds to the current preoccupations of researchers and policy-makers as regards the mitigation of the climate change risks.
Author Response
a) Literature Review. The current version of the manuscript includes the literature review in the section ”1. Introduction”. In order to underline the different viewpoints of researchers, a distinct section of the manuscript can be dedicated to the literature review, immediately after a revised ”1. Introduction”.
Response: Thank you very much for your valuable comments. Based on your suggestion, we have split the original introduction into two parts: Introduction and Literature Review. Please see lines 25-139.
b) Methodology: Research question(s). It would be useful to specify clearly the research question(s) at the beginning of the section ”Methodology and data”. Also, it would be necessary to explain the reasons for the use of the ARMAX modeling and of the Granger causality analysis in relation to the research question(s).
Response: We have made corresponding changes. Please see lines 141-147 and 185-187.
c) Implications for the research. The manuscript enumerates several policy implications. Nevertheless, the paper can benefit from the presentation of the implications for further research on the same topic. Examples of implications for future research can be provided.
Response: We have added feasibility suggestions for future research outlooks. Please see lines 493-502.
Reviewer 2 Report
Climate change is attacking ecosystems, reducing biodiversity and making it harder for many species to survive. It is also altering the storage of carbon in the cycle and fragmenting the habitats of each species. Such links also exist in terms of household energy consumption. Globally, there are numerous studies on the impact of consumption on the climate and vice versa. I would therefore be cautious in stating that such research is not being conducted (line 11).
The whole article demonstrates in a very statistical way the impact of climate change risks on residential consumption, but it should be taken into account that not every reader is a statistician, so I believe that the result should be more clearly and extensively described in the discussion of results which can contribute to the level of reading and citation of the article.
225 - 252 text formatting - interlineation - appears to be different from the rest of the text
In conclusion, I find the article itself very interesting and the research carried out extremely valuable. I wish you further success in continuing your research.
Author Response
Climate change is attacking ecosystems, reducing biodiversity and making it harder for many species to survive. It is also altering the storage of carbon in the cycle and fragmenting the habitats of each species. Such links also exist in terms of household energy consumption. Globally, there are numerous studies on the impact of consumption on the climate and vice versa. I would therefore be cautious in stating that such research is not being conducted (line 11).
Response: Thank you very much for your valuable comments. We have revised this expression from “there is no research on it” to “there is little research on it”. At the same time, we read the full text to avoid similar absolute statements. Please see lines 11 and 87-88.
The whole article demonstrates in a very statistical way the impact of climate change risks on residential consumption, but it should be taken into account that not every reader is a statistician, so I believe that the result should be more clearly and extensively described in the discussion of results which can contribute to the level of reading and citation of the article.
Response: We read the Empirical results and try to make the article more accessible to people in different fields. But truth be told, there is very little that can be changed.
225 - 252 text formatting - interlineation - appears to be different from the rest of the text.
Response: We have modified this problem and examined similar problems in full text and modified them.
In conclusion, I find the article itself very interesting and the research carried out extremely valuable. I wish you further success in continuing your research.
At the end, thank all the reviewers and editors again.